# Anchored Mixture-of-Experts for Efficient Time Series Forecasting

**Rui Wang**[1]   **Renhao Xue**[1]   **Ray Razi**[1]   **Huan Song**[1]   **Hannah R. Marlowe**[1]

## Abstract

Time series forecasting models are increasingly scaled through large Transformer backbones, yet most existing approaches process all series through a shared dense computation path despite substantial heterogeneity in temporal structure. Mixture-of-Experts (MoE) offers a natural alternative by enabling conditional computation, but standard MoE routing leaves expert specialization weakly identified and often unstable during downstream adaptation. We propose AME-TS, a regime-aware MoE framework for time series forecasting that aligns expert routing with interpretable temporal structure. AME-TS first uses a lightweight regime predictor to estimate series-level descriptors, including forecastability, seasonality, trend, and sparsity, and maps them to a soft structural prior over experts. This series-level prior guides token-level routing during training through a training-only prior-alignment loss, encouraging structure-aligned specialization. On the GIFT-Eval benchmark, AME-TS achieves state-of-the-art or competitive performance across model scales while using substantially fewer active parameters than recent time series foundation models. We further show that AME-TS learns more interpretable routing geometry and substantially more stable expert specialization than standard MoE during fine-tuning on M5. These results suggest that structure-aware routing is an effective and reliable way to realize the benefits of sparse expert models for time series forecasting.

## 1. Introduction

Time series forecasting is a fundamental problem in applications such as retail demand planning (4; 31), cloud operations (11; 26), healthcare (24), and weather pre-

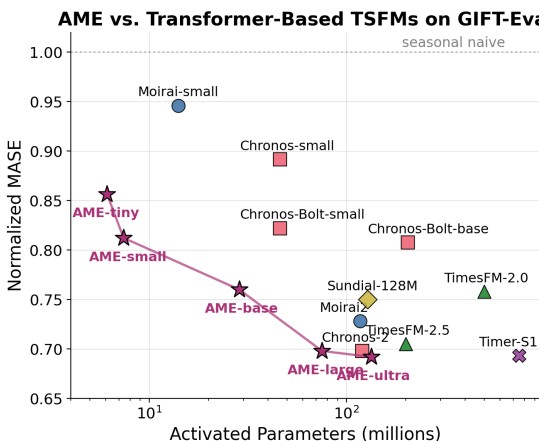

*Figure 1.* **MASE vs. activated parameter count on GIFT-Eval.** AME-TS achieves a favorable accuracy–efficiency tradeoff, matching or outperforming strong TSFMs while activating substantially fewer parameters.

diction (16; 30). Time series foundation models have achieved strong zero-shot and transfer performance by scaling architectures over large collections of time series (2; 6; 9; 3; 29; 34). However, most models process all series through a shared dense computation path, despite substantial heterogeneity in forecasting-relevant temporal structure.

Mixture-of-Experts (MoE) architectures (5) scale neural networks by conditionally activating only a subset of parameters for each input, and have been adopted successfully in both large language models (8; 15; 10; 27; 28) and recent time series forecasting models (18; 29). In practice, however, standard MoE routing leaves expert roles weakly identified: because experts are permutation-symmetric, specialization can be shaped by initialization, optimization, or fine-tuning dynamics rather than forecasting-relevant structure. Recent work further suggests that explicitly guided routing can improve MoE specialization, interpretability, and adaptation (13; 14; 33; 23; 28; 22).

Time series forecasting is a natural setting for structure-guided routing because properties such as forecastability (31; 12), seasonality, trend, and sparsity provide interpretable axes for organizing expert specialization. Motivated by this view, we propose Anchored Mixture-of-Experts (AME-TS), a structure-guided sparse time series foundation model. AME-TS uses a lightweight regime predictor

[1]Amazon Web Services, USA. Correspondence to: Rui Wang <rwngamz@amazon.com>.

*Proceedings of the $2^{nd}$ ICML Workshop on Foundation Models for Structured Data*, Seoul, South Korea. 2026. Copyright 2026 by the author(s).

to estimate temporal descriptors of each input series, maps them to a series-level soft prior over experts, and applies a prior-alignment loss to guide token-level routing during training. This encourages structure-aligned specialization while preserving flexible learned routing at inference time. The core idea is to use temporal descriptors not as hard routing rules, but as a training-only structural prior that breaks expert symmetry and biases sparse capacity toward interpretable forecasting regimes. Our contributions are summarized as follows.

- We propose AME-TS, a structure-guided sparse time series foundation model that constructs a soft prior over expert usage from interpretable temporal descriptors and introduces a prior-alignment loss to align token-level MoE routing with series-level temporal structure.

- On GIFT-Eval (1), AME-TS achieves a strong accuracy–efficiency tradeoff, as summarized in Figure 1, outperforming small-scale TSFMs and remaining competitive with larger baselines while using fewer active parameters.

- AME-TS learns interpretable routing and representation spaces organized around meaningful temporal regimes.

- On the M5 dataset (21), AME-TS achieves strong zero-shot and fine-tuned performance while maintaining substantially more stable expert specialization than standard MoE during fine-tuning.

## 2. Methodology

### 2.1. AME-TS Overview

AME-TS is a structure-guided sparse MoE forecasting model that aligns expert specialization with interpretable temporal structure. As illustrated in Figure 2, AME-TS balances a series-level structural prior with token-level learned routing. The series-level prior summarizes global temporal properties of the input series, while the token-level router selects experts from local representations inside the forecasting backbone. A lightweight regime predictor estimates a soft structural profile consisting of forecastability, seasonality strength, trend strength, and sparsity. This profile is mapped to a soft prior over experts, biasing expert specialization toward interpretable temporal regimes. We use an encoder-only Transformer forecasting backbone with patchified inputs and masked prediction over the forecast horizon, replacing dense feed-forward layers with MoE layers; implementation details and the full training objective are provided in Appendix A.3.

### 2.2. Structural Profile and Expert Prior

**Structural descriptors.** For a univariate input window $X = (x_1, \ldots, x_T)$, we compute four complementary struc-

tural descriptors: forecastability, seasonality strength, trend strength, and sparsity. Forecastability measures spectral concentration via normalized power spectral entropy (31; 12); seasonality strength measures the variation explained by the STL seasonal component (7); trend strength measures normalized linear change; and sparsity measures repeated-value or intermittent behavior. These descriptors provide a compact, non-redundant structural profile for guiding expert specialization. Full definitions and correlation analysis are provided in Appendix A.4.

**Regime predictor.** Computing descriptors analytically for every training sample is prohibitively expensive. We therefore compute them on a small subset of the pretraining pool and train a lightweight regime predictor $g_\phi$ to provide fast estimates:

$$g_\phi(X) = [r_{\mathrm{f}}, r_{\mathrm{s}}, r_{\mathrm{t}}, r_{\mathrm{sp}}] \in [0, 1]^4,$$

where $r_{\mathrm{f}}, r_{\mathrm{s}}, r_{\mathrm{t}}, r_{\mathrm{sp}}$ denote forecastability, seasonality strength, trend strength, and sparsity scores, respectively. For multivariate inputs, descriptors are computed per variate. More details are provided in Appendix A.5.

**Specialized and shared experts.** We partition experts into specialized experts $\mathcal{E}_{\mathrm{sp}}$, anchored to the $D = 4$ structural descriptors, and shared experts $\mathcal{E}_{\mathrm{sh}}$, which handle weak or ambiguous structural profiles. Let $q_{\mathrm{anchor}}(e \mid d)$ assign descriptor $d$ to a subset of specialized experts. The specialized prior is

$$q_{\mathrm{sp}}(e \mid X) \propto \sum_{d=1}^{D} q_{\mathrm{anchor}}(e \mid d)\, g_\phi(d \mid X), \quad e \in \mathcal{E}_{\mathrm{sp}},$$

with normalization over $\mathcal{E}_{\mathrm{sp}}$. To allocate mass to shared experts, define

$$H(X) = \frac{1}{D} \sum_{d=1}^{D} h(g_\phi(d \mid X)), \quad S(X) = \max_d g_\phi(d \mid X),$$

where $h(\cdot)$ is binary entropy. We define

$$\pi_{\mathrm{sh}}(X) = \big(1 - S(X)\big)\sigma(\alpha H(X) - b),$$

where $\sigma$ is the sigmoid function and $\alpha, b$ are hyperparameters or learnable scalars. The final prior is

$$q(e \mid X) = \begin{cases} \pi_{\mathrm{sh}}(X)/|\mathcal{E}_{\mathrm{sh}}|, & e \in \mathcal{E}_{\mathrm{sh}}, \\ (1 - \pi_{\mathrm{sh}}(X))q_{\mathrm{sp}}(e \mid X), & e \in \mathcal{E}_{\mathrm{sp}}. \end{cases}$$

This soft prior breaks the permutation symmetry of standard MoE routing by anchoring expert preferences to meaningful regimes, while still avoiding hard expert assignments.

### 2.3. Training with Prior Alignment

The structural prior $q(e \mid X)$ is defined at the series level, while MoE routing operates at the token level. For the

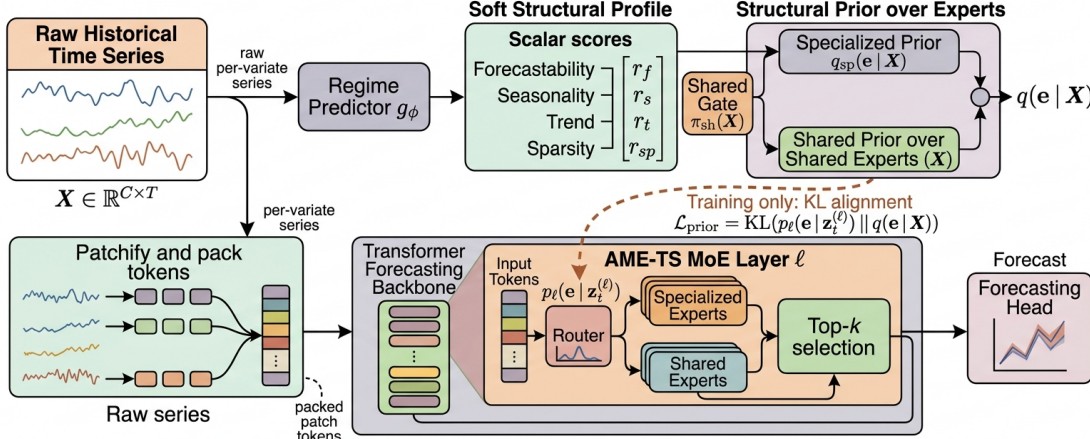

*Figure 2.* Overview of AME-TS. A regime predictor extracts a soft structural profile from raw time series to construct a structural prior over experts $q(e \mid X)$, while patchified tokens are processed by a Transformer forecasting backbone with AME-TS MoE layers. Training uses KL alignment between token-level routing and the structural prior; inference uses only the learned router.

representation $z_t^{(\ell)}$ of patch token $t$ at layer $\ell$, the router produces $p_\ell(e \mid z_t^{(\ell)}) = \mathrm{softmax}(W_r^{(\ell)} z_t^{(\ell)})$. We align token-level routing with the series-level prior using

$$\mathcal{L}_{\mathrm{prior}} = \frac{1}{N_L} \sum_{\ell=0}^{N_L-1} \lambda_\ell \mathbb{E}_t \Big[ \mathrm{KL}\big(p_\ell(e \mid z_t^{(\ell)}) \,\|\, q(e \mid X)\big) \Big],$$

where $N_L$ is the number of layers and $\lambda_\ell = \lambda_{\max}\ell/(N_L - 1)$. This training-only alignment transfers interpretable series-level structure into token-level sparse routing while leaving inference to the learned router.

## 3. Experiment

### 3.1. Experimental Setup

We pre-train AME-TS on a heterogeneous pool of 96 dataset configurations across 8 domains, containing 3.5M time series and 18B observations with frequencies from seconds to yearly; details are provided in Appendix A.6. We train with AdamW, linear warmup, and linear learning-rate decay on one p4 instance with 8 A100 GPUs. Following the official GIFT-Eval protocol (1), we report MASE, sMAPE, RMSE, and MAE over 97 tasks. Architecture configurations and training details are provided in Appendix A.3 and A.7.

### 3.2. GIFT-Eval Results and Ablation

We evaluate AME-TS on GIFT-Eval (1), which contains 97 forecasting tasks across diverse datasets, frequencies, and horizons. We compare against the top published time series foundation models from the public leaderboard across multiple parameter scales. Since Moirai-MoE (18) and TimeMoE (29) do not appear as directly comparable named entries on the public leaderboard, our main comparison follows the listed TSFM baselines.

Figure 1 summarizes the main accuracy–efficiency comparison in terms of MASE versus activated parameter count. AME-TS achieves the best MASE among the listed models, with AME-TS Ultra reaching 0.692 while activating 133M parameters per token. The gains are especially pronounced at small scales: AME-TS Small outperforms both Moirai-Small and Chronos-Small while activating only 7M parameters per token. Full GIFT-Eval results across MASE, sMAPE, RMSE, and MAE, including task-specific per-dataset comparisons, are provided in Appendix Table 1.

We further conduct ablations to isolate the contribution of routing design. As shown in Appendix Table 2, sparse capacity alone improves over the dense baseline, but structure-guided routing provides additional gains; removing any single descriptor also degrades performance, confirming that the four structural descriptors provide complementary routing signals.

### 3.3. Routing Interpretability

We evaluate whether AME-TS learns routing patterns and internal representations that align with interpretable temporal structure. We compare AME-TS against a standard MoE with the same backbone and expert architecture but without structure-guided routing. From the same layer in both models, we extract router logits, encoder representations, top-1 expert assignments, and structural labels derived from the regime predictor.

Figure 3 visualizes router space and encoder space using t-SNE. In router space, points are colored by structural labels derived from forecastability and sparsity; in encoder space, points are colored by top-1 expert assignment. AME-TS exhibits clearer structure-aligned grouping in router space and much stronger expert separation in encoder space than

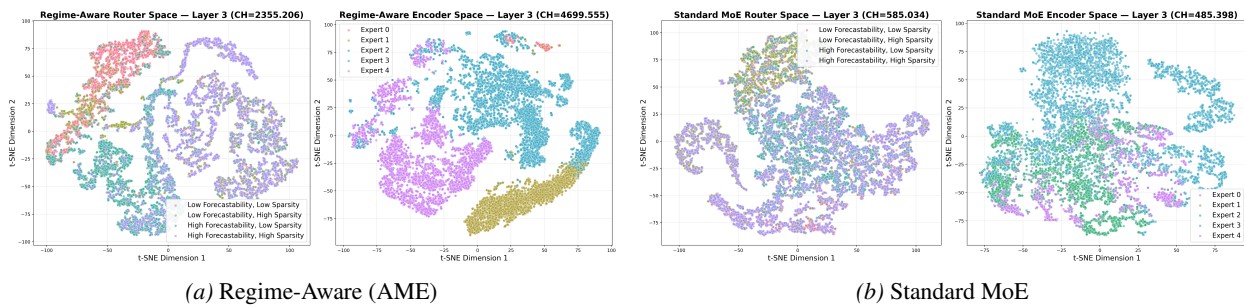

*(a)* Regime-Aware (AME)          *(b)* Standard MoE

*Figure 3.* t-SNE visualizations comparing AME-TS and standard MoE at the same layer. Each subfigure contains two panels: router space on the left and encoder space on the right. In the router space, points are colored by regime profiles derived from forecastability and sparsity labels. In the encoder space, points are colored by top-1 expert assignment. AME-TS yields clearer regime-aligned structure in router space and substantially stronger expert separation in encoder space, as also reflected by the higher Calinski–Harabasz (CH) scores reported in the subplot titles.

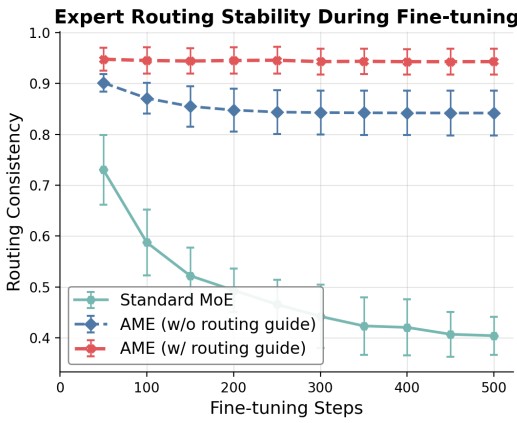

*Figure 4.* **Routing stability during fine-tuning on M5.** AME-TS maintains substantially more stable expert specialization than standard MoE, and routing guidance further improves stability during adaptation.

standard MoE. This is also reflected by higher Calinski–Harabasz scores, indicating that the structural prior induces a more interpretable routing geometry and more specialized expert representations. Additional visualization and metric details are provided in Appendix A.9.

### 3.4. Transfer and Fine-Tuning on M5

We evaluate AME-TS on the M5 Walmart dataset (21), which is not included in pre-training. M5 contains 30,490 daily retail time series across 12 hierarchical levels and is evaluated using WRMSSE over a 28-day horizon. We report both zero-shot and fine-tuned performance, using the first-place M5 competition result as a task-specific reference.

Even without task-specific adaptation, AME-TS outperforms the competition winner on all three item-level aggregations (Prod, Prod-St, and Prod-Str), suggesting that structure-guided routing transfers well to an unseen retail dataset with diverse and sparse temporal patterns. After fine-

tuning, AME-TS achieves an average WRMSSE of 0.506, improving over the first-place M5 result (0.520), with the strongest gains again appearing at lower hierarchical levels. Full results across all 12 hierarchy levels are provided in Appendix Table 6.

### 3.5. Routing Stability During Fine-Tuning

We evaluate whether expert specialization remains stable during fine-tuning on M5. We compare AME-TS against two baselines with the same architecture: standard MoE and an ablated variant of AME-TS in which routing guidance is removed during fine-tuning. We measure routing consistency as the agreement between current top-1 expert assignments and those at the initial checkpoint on a fixed probe set of 1,000 series sampled across M5 hierarchy levels. Full metric details are provided in Appendix A.11.

Figure 4 shows that AME-TS maintains substantially higher routing consistency throughout fine-tuning than standard MoE. The ablated variant without routing guidance remains relatively stable, suggesting that AME-TS already learns meaningful expert structure during pre-training, while standard MoE exhibits substantial drift. These results indicate that structure-guided routing produces expert specialization that remains coherent during adaptation.

## 4. Discussion

AME-TS shows that aligning MoE routing with temporal structure can improve forecasting efficiency, interpretability, and adaptation stability. Across GIFT-Eval and M5, the results suggest that sparse capacity is most effective when expert specialization is guided by forecasting-relevant structure rather than learned purely from routing dynamics. A limitation of the current work is that AME-TS focuses on historical time series inputs; future dynamics may also depend on external context such as text, events, or metadata. Extending structure-aware routing to multimodal forecasting is therefore a natural next direction.

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

# A. Additional Experimental and Implementation Details

## A.1. Full GIFT-Eval Results

Table 1 reports the complete GIFT-Eval comparison across MASE, sMAPE, RMSE, and MAE, including both foundation-model and task-specific per-dataset settings. This table supports the accuracy–efficiency trend summarized in Figure 1.

## A.2. Ablation Study

Additional ablations are reported in Table 2. Replacing a dense model with standard MoE yields modest improvements, confirming that sparse expert capacity alone is beneficial. Introducing regime-aware routing yields further gains, showing that the improvement comes not only from increased capacity, but from better alignment between routing decisions and temporal structure. Among prior-integration strategies, KL-guided routing consistently outperforms additive prior injection, indicating that soft alignment is more effective than direct prior injection. Forward and reverse KL achieve similar performance, with forward KL showing slightly more consistent gains across tasks.

We further study the contribution of individual structural descriptors by dropping one descriptor at a time. Removing any single descriptor degrades performance, confirming that the four structural descriptors provide complementary routing signals. The largest drops are observed when forecastability or sparsity is removed, suggesting that these features provide especially informative cues for expert allocation. Taken together, these ablations show that the gains of AME-TS arise from structure-aligned routing rather than sparse capacity alone.

## A.3. Model Architecture Details

We evaluate five AME-TS variants at different scales: Tiny, Small, Base, Large, and Ultra. Their detailed configurations are reported in Table 4. In the dataset-specific setting, model size is selected separately for each dataset and is generally much smaller than in the foundation-model setting.

**Forecasting and training objective.** The primary forecasting objective $\mathcal{L}_{\text{task}}$ is a masked prediction loss over the forecast horizon. Given predictions $\hat{y}$ and ground-truth targets $y$, we use an $\ell_1$ loss:

$$\mathcal{L}_{\text{task}} = \mathbb{E}\left[\|\hat{y} - y\|_1\right].$$

The final training objective combines the forecasting loss with the prior-alignment and orthogonality losses:

$$\mathcal{L} = \mathcal{L}_{\text{task}} + \lambda_{\text{prior}}\mathcal{L}_{\text{prior}} + \lambda_{\text{ortho}}\mathcal{L}_{\text{ortho}}.$$

When multiple experts are associated with the same descriptor, we include an orthogonality loss to promote diversity among their outputs:

$$\mathcal{L}_{\text{ortho}} = \mathbb{E}_{i \neq j}\left[|\langle h_i, h_j \rangle|\right],$$

where $h_i$ and $h_j$ are outputs of co-activated experts within the same descriptor group.

## A.4. Structural Descriptor Definitions

We compute four structural descriptors for each input series: forecastability, seasonality strength, trend strength, and sparsity. These descriptors provide complementary signals for constructing the structural prior used in AME-TS.

**Forecastability.** Forecastability measures how predictable a series is from its frequency-domain structure. Following prior work (31; 12), we quantify it using the entropy of the normalized power spectral density. Given an input series $X = (x_0, x_1, \ldots, x_{T-1})$, let $\widetilde{X} = \text{Detrend}(X)$ denote its detrended version, and let $p_i$ denote the normalized power of the $i$-th frequency bin of $\widetilde{X}$. The spectral entropy is

$$H_a(\widetilde{X}) = -\sum_i p_i \log_a p_i,$$

and the forecastability score is

$$\text{Forecastability}(X) = 1 - \frac{H_a(\widetilde{X})}{\log_a N_f},$$

where $N_f$ is the number of frequency bins.

**Seasonality strength.** We quantify seasonality strength using STL decomposition (7). Each series is decomposed into trend, seasonal, and remainder components, where the seasonal period is determined from the dominant Fast Fourier Transform peak frequency. Let $S$ and $R$ denote the seasonal and residual components, respectively. We define

$$\text{SeasonalityStrength}(X) = 1 - \frac{\text{Var}(R)}{\text{Var}(S + R)}.$$

**Trend strength.** Trend strength measures the magnitude of linear change across the input window. We first min–max normalize the series to $[0, 1]$, fit a linear regression, and let $\hat{\beta}$ denote the fitted slope. We define

$$\text{TrendStrength}(X) = \min(1, |\hat{\beta}|T).$$

This measures the directional change over the input window relative to the series range.

*Table 1.* Forecasting performance on GIFT-Eval (97 tasks). Scores are the geometric mean of each metric normalized by the Seasonal Naive baseline (lower is better). Best results within each scale group are shown in **bold**. [†] Active parameters per token due to sparse MoE routing.

| Scale | Model | #Params | MASE ↓ | sMAPE ↓ | RMSE ↓ | MAE ↓ |
|---|---|---|---|---|---|---|
| Small | Moirai-Small (34) | 14M | 0.946 | 1.109 | 1.056 | 0.906 |
| | Chronos-Small (3) | 46M | 0.892 | 1.059 | 0.866 | 0.847 |
| | **AME-TS Tiny** | **14M[†]6M** | 0.856 | 1.043 | 0.808 | 0.834 |
| | **AME-TS Small** | **30M[†]7M** | 0.812 | 0.990 | 0.767 | 0.741 |
| Base | Chronos-2 (2) | 120M | 0.698 | 0.886 | 0.704 | 0.679 |
| | Sundial-128M (20) | 128M | 0.750 | 0.945 | 0.739 | 0.739 |
| | **AME-TS Base** | **62M[†]29M** | 0.765 | 0.926 | 0.737 | 0.725 |
| Large | TimesFM-2.5 (9) | 231M | 0.705 | 0.856 | 0.694 | 0.687 |
| | Moirai2 (17) | 300M | 0.728 | 0.925 | 0.739 | 0.705 |
| | **AME-TS Large** | **162M[†]75M** | 0.700 | 0.852 | 0.690 | 0.680 |
| | **AME-TS Ultra** | **540M[†]133M** | 0.692 | 0.857 | 0.687 | 0.680 |
| Per-dataset | PatchTST (25) | – | 0.849 | 1.020 | 0.839 | 0.810 |
| | iTransformer (19) | – | 0.893 | 1.074 | 0.858 | 0.850 |
| | **AME-TS** | – | 0.844 | 0.946 | 0.832 | 0.800 |

*Table 2.* Ablation study of AME-TS Tiny on GIFT-Eval benchmark.

| Configuration | MASE ↓ |
|---|---|
| *(a) Architecture* | |
| **AME-TS Tiny (proposed)** | **0.856** |
| Standard MoE (no prior) | 0.929 |
| Dense (no MoE) | 0.958 |
| *(b) Prior integration* | |
| Reverse KL | 0.917 |
| Additive prior | 1.060 |
| *(c) Regime ablation* | |
| Drop Forecastability | 0.919 |
| Drop Seasonality | 0.889 |
| Drop Trend | 0.882 |
| Drop Sparsity | 0.910 |

**Sparsity.** Sparsity captures how intermittent or repeated-value dominated a series is. We define

$$\text{Sparsity}(X) = 1 - \frac{N_{\text{unique}}(X)}{T},$$

where $N_{\text{unique}}(X)$ is the number of unique values in the input window.

### A.5. Regime Predictor Architecture and Training

The regime predictor $g_\phi$ is trained separately from the forecasting model and kept frozen during AME-TS training. Its role is to map a raw input series to a soft structural profile over the four structural descriptors: forecastability, seasonality strength, trend strength, and sparsity. Because these properties are not mutually exclusive, $g_\phi$ predicts them independently rather than assigning each series to a single discrete class. The resulting output is a soft structural pro-

file that summarizes the forecasting-relevant characteristics of the input series.

The predictor is trained on a subset of the same pre-training pool used for the forecasting model. Computing the analytical structural descriptors for every series in the full pre-training pool would be prohibitively expensive, so we instead construct a sampled training set by drawing approximately 98,910 random crops. For each crop, ground-truth structural descriptor targets are computed using the analytical feature pipeline described in Section 2.2.

Rather than using a single multi-output network, we train four separate single-feature predictors, one for each structural descriptor: forecastability, seasonality strength, trend strength, and sparsity. Each predictor has approximately 450K parameters, for a total of roughly 1.8M parameters across all four predictors.

The architecture consists of a multi-scale 1D CNN encoder with three parallel branches using kernel sizes 5, 11, and 21. Each branch contains two Conv1d layers with GroupNorm, GELU activations, and MaxPool(2). In parallel, we apply a self-attention block to the first-scale feature map, consisting of LayerNorm, 4-head MultiheadAttention, and mean pooling. The pooled outputs from the three CNN branches and the attention block are concatenated into a 512-dimensional representation, which is fed to an MLP head of the form $512 \rightarrow 128 \rightarrow 64 \rightarrow 1$, with GELU activations, dropout of 0.1 after the first hidden layer, and a sigmoid output in $[0, 1]$.

The four structural descriptors exhibit heterogeneous label distributions. For example, forecastability is often concentrated in a relatively narrow range, whereas sparsity spans a much broader portion of $[0, 1]$. Training directly on raw

targets can therefore bias the predictors toward densely populated regions of the label distribution. To mitigate this, we apply per-feature quantile normalization to the structural descriptor targets before training. For each feature, target values are mapped to their empirical ranks within the sampled training pool, yielding approximately uniform targets on $[0, 1]$. This encourages the predictor to use the full target range rather than focusing disproportionately on highly concentrated regions. The same training quantiles are used to normalize validation targets.

We train the regime predictors using mean squared error loss on the quantile-normalized targets. Optimization uses AdamW with learning rate $10^{-3}$, weight decay $10^{-4}$, cosine annealing for 100 epochs, batch size 256, and early stopping with patience 15 based on validation MSE.

### A.6. Pre-training Data Composition

We pre-train AME-TS on a heterogeneous corpus assembled from three public sources. First, we draw a subset of datasets from the LOTSA archive (34), which contains more than 170 datasets totaling 27 billion observations; our subset represents less than 20% of the full archive and spans traffic, weather, energy, retail, healthcare, web/cloud, and economics. Second, we incorporate several unique datasets from the Chronos pre-training corpus (3), including a one-million-series synthetic KernelSynth corpus and several real-world datasets not otherwise available in LOTSA. Third, we generate a synthetic dataset of 87,000 time series from diverse parametric models, including seasonal, trend, sparse, and noise processes, with known structural descriptor labels across four dimensions (forecastability, seasonality, trend, sparsity) to encourage expert specialization during routing. We explicitly exclude the M5 retail hierarchy from pre-training, reserving it for downstream fine-tuning and evaluation, and we verify that our corpus contains no overlap with the held-out test windows of standard forecasting benchmarks, so all reported results are free of test-data leakage. In total, the pre-training pool spans 96 dataset configurations across 8 domains (Table 3), comprising approximately 3.5 million individual time series and 18 billion observations, with frequencies ranging from 10-second to yearly. We use domain-balanced sampling with per-dataset weights to prevent large datasets (e.g., buildings_900k, kernel_synth_1m) from dominating training.

### A.7. Additional Training Hyperparameters

For pre-training, we optimize AME-TS using AdamW with $\beta_1 = 0.9$, $\beta_2 = 0.98$, and weight decay 0.01. We use a peak learning rate of $5 \times 10^{-4}$ with a cosine-with-restarts schedule over three cycles and a warmup of 5,000 steps. Training runs for 200 epochs with 2,000 steps per epoch, for a total of 400K optimization steps. We use a batch size

*Table 3.* Pre-training data composition (96 configurations, 8 domains).

| Domain | #Datasets | #Series | #Obs. |
|---|---|---|---|
| Econ/Fin | 15 | 104K | 28M |
| Energy | 23 | 1.80M | 16.16B |
| Transport | 16 | 5K | 148M |
| Weather/Climate | 17 | 70K | 65M |
| Sales/Retail | 7 | 246K | 227M |
| Healthcare | 6 | 1K | 0.2M |
| Web/Cloud | 10 | 207K | 371M |
| Synthetic | 2 | 1.09M | 1.04B |
| **Total** | **96** | **3.52M** | **18.04B** |

of 32 per GPU on 8 GPUs, giving an effective batch size of 256, and apply gradient clipping with maximum norm 2.0. During masked forecasting pre-training, 15%–50% of input tokens are randomly masked for prediction. The maximum sequence length is 512 tokens, training uses TF32 precision, and the regime predictor is kept frozen, with its input length capped at 192 timesteps.

For fine-tuning on M5, we use a substantially smaller learning rate of $10^{-5}$ and no learning-rate schedule. The batch size is set between 8 and 16 depending on the configuration, and models are fine-tuned for 6,000–10,000 steps. We apply level-balanced sampling so that each M5 hierarchy level is sampled with equal probability during training. The regime predictor remains active, and the KL alignment loss uses the same coefficient as in pre-training.

### A.8. Feature Correlation Analysis

To assess whether the proposed structural descriptors provide complementary information, Table 5 reports their Pearson correlation matrix on 98,910 training samples from the pre-training pool. The off-diagonal correlations are uniformly moderate in magnitude, with the largest absolute correlation equal to 0.296. These results suggest that forecastability, seasonality, trend, and sparsity capture distinct temporal properties and therefore provide complementary, non-redundant signals for both the regime predictor and the routing prior.

*Table 4.* Architecture details of AME-TS variants.

| Model | $d$ | $L$ | $E/k$ | Params | Active |
|---|---|---|---|---|---|
| tiny | 256 | 4 | 5/2 | 14M | 6M |
| small | 256 | 5 | 10/2 | 30M | 7M |
| base | 512 | 5 | 5/2 | 62M | 29M |
| large | 768 | 6 | 5/2 | 162M | 75M |
| ultra | 1024 | 6 | 10/2 | 540M | 133M |

*Table 5.* Pearson correlation between the four structural descriptors across 98,910 training samples. Moderate correlations ($|r| < 0.5$) indicate that the features capture distinct temporal properties.

|  | **Forecast.** | **Season.** | **Trend** | **Sparsity** |
|---|---|---|---|---|
| Forecastability | 1.000 | 0.111 | 0.200 | −0.296 |
| Seasonality | 0.111 | 1.000 | −0.280 | −0.031 |
| Trend | 0.200 | −0.280 | 1.000 | −0.277 |
| Sparsity | −0.296 | −0.031 | −0.277 | 1.000 |

## A.9. Routing Interpretability Details

We evaluate whether AME-TS learns routing patterns and internal representations that align with interpretable temporal structure. We compare AME-TS against a standard MoE model with the same backbone and expert architecture but without structure-guided routing. From the same layer in both models, we extract router logits as routing embeddings, encoder representations, top-1 expert assignments, and structural descriptor predictions from the regime predictor.

To visualize the learned spaces, we project both routing embeddings and encoder representations into two dimensions using t-SNE. In the router space, points are colored by structural profiles obtained by thresholding the regime predictor outputs. In Figure 3, we focus on binary labels derived from forecastability and sparsity, which yields four structural groups. In encoder space, points are colored by top-1 expert assignment. This lets us separately examine whether routing geometry aligns with temporal structure and whether encoder representations organize according to expert specialization.

Figure 3 shows a clear qualitative difference between AME-TS and standard MoE. In the router space, AME-TS forms more coherent regions associated with different structural labels, whereas standard MoE exhibits substantially greater mixing across groups. In the encoder space, AME-TS produces sharply separated expert-specific regions, while standard MoE yields weaker and more entangled clusters. These patterns suggest that the structural prior not only affects routing decisions directly, but also reshapes the representation geometry learned by the backbone.

We quantify this effect using the Calinski–Harabasz (CH) index (32), which measures cluster compactness and separation. Higher CH values indicate better-defined clusters. As reported in the subplot titles of Figure 3, AME-TS achieves substantially higher CH scores than standard MoE in both router and encoder space, indicating more structured routing behavior and stronger expert specialization.

## A.10. M5 Transfer Results

We evaluate AME-TS-Base on the M5 Walmart dataset (21), which is not included in pre-training. M5 contains 30,490 daily retail time series organized into 12 hierarchical levels and is evaluated using WRMSSE over a 28-day forecasting horizon. Because many recent forecasting foundation models include M5 in pre-training, we avoid direct zero-shot comparison to such models and instead report results against the first-place M5 competition result as a task-specific reference.

Table 6 reports results across all 12 hierarchy levels. Even without task-specific adaptation, AME-TS outperforms the competition winner on all three item-level aggregations (Prod, Prod-St, Prod-Str). This is notable because lower-level M5 series are more heterogeneous and often exhibit higher sparsity and less regular temporal structure.

We then fine-tune AME-TS on M5 training data, incorporating a day-of-week calendar covariate as a second input variate. Because the model treats each variate as a separate token sequence sharing the same time index, the day-of-week signal is fully observed at both context and forecast positions. Fine-tuned AME-TS achieves an average WRMSSE of 0.506, improving over the first-place M5 result of 0.520. The gains are strongest at the item level, where AME-TS reduces WRMSSE by 13–22% relative to Rank 1. Fine-tuned AME-TS still falls short of Rank 1 on some higher aggregation levels, where strong seasonality and hierarchical reconciliation favor heavily engineered task-specific pipelines.

## A.11. Routing Stability Details

We evaluate whether expert specialization remains stable during fine-tuning on M5. We compare AME-TS against two baselines with identical architectures: standard MoE (no prior) and an ablated variant of AME-TS in which routing guidance is removed during fine-tuning.

We define routing consistency as the agreement between current routing decisions and those at the initial checkpoint. For a fixed probe set $\mathcal{D}_{\text{probe}}$, let $e_{i,t,\ell}^{(0)}$ and $e_{i,t,\ell}^{(k)}$ denote the top-1 expert assigned to token $t$ of series $i$ at layer $\ell$ at the initial and $k$-th fine-tuning step, respectively. We compute

$$\text{RC}(k) = \frac{1}{|\mathcal{S}|} \sum_{(i,t,\ell) \in \mathcal{S}} \mathbf{1}\left[e_{i,t,\ell}^{(k)} = e_{i,t,\ell}^{(0)}\right],$$

where $\mathcal{S}$ denotes the set of tracked series-token-layer tuples derived from $\mathcal{D}_{\text{probe}}$. Higher values indicate more stable expert specialization. We construct a fixed probe set of 1,000 time series sampled across all hierarchy levels of M5 to capture diverse temporal behaviors.

Figure 4 shows routing consistency over fine-tuning steps.

*Table 6.* WRMSSE on the M5 dataset across all 12 hierarchical levels. Lower is better. "Rank 1" denotes the first-place M5 competition result, reported here as a task-specific reference.

| Model | Tot | St | Str | Cat | Dept | St-Cat | St-Dept | Str-Cat | Str-Dept | Prod | Prod-St | Prod-Str | Average |
|---|---|---|---|---|---|---|---|---|---|---|---|---|---|
| M5 Rank 1 (21) | 0.199 | 0.310 | 0.400 | 0.277 | 0.365 | 0.390 | 0.474 | 0.480 | 0.573 | 0.966 | 0.929 | 0.884 | 0.520 |
| AME-TS (Zero-shot) | 0.367 | 0.470 | 0.536 | 0.424 | 0.553 | 0.517 | 0.620 | 0.606 | 0.684 | 0.851 | 0.860 | 0.867 | 0.613 |
| AME-TS (Fine-tuned) | 0.226 | 0.357 | 0.430 | 0.318 | 0.441 | 0.419 | 0.521 | 0.502 | 0.579 | 0.754 | 0.781 | 0.765 | **0.506** |

Confidence intervals are computed over five independent fine-tuning runs with learning rates sampled from $[1 \times 10^{-5}, 5 \times 10^{-5}]$. AME-TS with structure-guided routing maintains consistently high stability throughout training. The ablated variant without routing guidance also remains relatively stable, suggesting that AME-TS learns meaningful expert specialization during pre-training, whereas standard MoE exhibits substantial routing drift.

