# OpenReview forum: "AME-TS: Anchored Mixture-of-Experts for Time Series Forecasting"
_ICML.cc/2026/Workshop/FMSD — FMSD @ ICML 2026 Poster_

### Official Review · Reviewer_VfLx · 2026-05-16

**Rating:** 8
**Confidence:** 4

**Review:**

Summary of contributions

This paper proposes AME-TS, a structure-guided Mixture-of-Experts framework for time series forecasting. The method introduces a lightweight regime predictor that estimates interpretable temporal descriptors, including forecastability, seasonality, trend strength, and sparsity, and maps them into a soft prior over experts. A KL-based prior alignment objective is further used to encourage token-level MoE routing to align with the series-level structural prior during training. Experimental results on GIFT-Eval and M5 demonstrate competitive forecasting performance and improved routing stability compared with standard MoE-based forecasting models.

Strengths
1. Well-motivated problem formulation. The paper addresses an important limitation of existing TSFMs and MoE forecasting models, namely that expert specialization is often weakly identified and unstable. The motivation for incorporating interpretable temporal structure into routing is intuitive and well aligned with classical forecasting principles.
2. Elegant integration of temporal structure and sparse routing. The proposed structural prior based on forecastability, seasonality, trend, and sparsity provides a simple yet effective mechanism for guiding expert specialization. The idea of using descriptor-based priors as training-only routing guidance, rather than hard assignments, is technically appealing.
3. Strong interpretability and routing stability analysis. Beyond forecasting accuracy, the paper provides useful analyses of routing behavior through t-SNE visualization and routing consistency evaluation during fine-tuning. The results suggest that AME-TS learns more coherent and stable expert specialization than standard MoE architectures.

Weaknesses
1. The structural descriptors remain relatively hand-crafted and limited. The proposed routing prior relies on four manually designed descriptors, which may not fully capture more complex temporal dynamics such as long-range interactions or cross-variable relationships.
2. The expert anchoring mechanism appears somewhat heuristic. The mapping between descriptors and expert groups through q_{anchor}(e|d) seems largely predefined, yet the paper does not analyze the sensitivity of performance to different anchor allocations or expert partition strategies.
3. The efficiency claims could be discussed more carefully. Although AME-TS achieves favorable performance with fewer activated parameters than dense TSFMs, it is unclear how much of the gain comes specifically from the proposed anchored routing mechanism versus the MoE architecture itself. In addition, the proposed framework introduces extra components, including descriptor computation and prior-alignment training, which may increase training complexity and overhead.

Suggestions
1. It would strengthen the paper to explore richer or learnable structural priors beyond the current manually designed descriptor space, potentially incorporating more complex temporal characteristics or adaptive regime representations.
2. Additional ablation studies on expert anchoring strategies and expert partition sensitivity would help clarify the robustness and scalability of the proposed routing design.
3. The interpretability analysis could be further improved with more quantitative metrics, such as expert specialization entropy, routing sparsity statistics, or mutual information between temporal descriptors and expert assignments.

---

### Official Review · Reviewer_Ri1m · 2026-05-20
**Review for AME**

**Rating:** 7
**Confidence:** 4

**Review:**

Summary:
This paper proposes a regime-aware MoE framework for time series forecasting, which aligns expert routing with interpretable temporal structure. Specifically, it first uses a lightweight regime predictor to estimate series-level descriptors, and then the series-level prior guides token-level routing during training, encouraging structure-aligned specialization.

Strengths:
1. The proposed AME-TS achieves SOTA or competitive performance across model scales while using substantially fewer active parameters on the GIFT-Eval benchmark.
2. AME-TS is also shown to gain more interpretable routing geometry and substantially more stable expert specialization than standard MoE during fine-tuning on M5.
3. The proposed AME-TS is adequately compared with the latest TSFMs.

Weaknesses:
1. It would be better if the training and fine-tuning cost is reported and compared with SOTA models.

---

### Official Review · Reviewer_HZdc · 2026-05-22
**Review for submission 56**

**Rating:** 8
**Confidence:** 4

**Review:**

Review:

The paper proposes AME-TS, a structure-guided Mixture-of-Experts framework for efficient time series forecasting. The main idea is to use interpretable temporal descriptors, including forecastability, seasonality, trend, and sparsity, to construct a soft series-level prior over experts. This prior guides token-level MoE routing during training through a KL prior-alignment loss, while inference uses only the learned router. Experiments on GIFT-Eval show that AME-TS achieves strong accuracy-efficiency tradeoffs, with AME-TS Ultra reaching MASE (=0.692) while activating 133M parameters per token. Additional M5 experiments show strong transfer and more stable expert specialization during fine-tuning compared with standard MoE.

Strengths:

(1) The paper addresses an important problem in time series foundation models: how to make sparse expert models more efficient, interpretable, and stable for heterogeneous time series data.

(2) The method is well motivated. Standard MoE routing can suffer from weak expert specialization and unstable routing, and using time-series structure as a soft routing prior is a natural and interesting solution.

(3) The empirical results are strong. AME-TS achieves competitive or state-of-the-art performance on GIFT-Eval while activating fewer parameters than several dense or larger time series foundation model baselines.

(4) The paper includes useful analysis beyond forecasting metrics, including ablations, routing-space visualization, and routing stability during M5 fine-tuning. These results support the claim that AME-TS learns more interpretable and stable expert specialization than standard MoE.

Areas for Improvement:

(1) The experimental scope is still somewhat limited. The main results rely heavily on GIFT-Eval and M5, so it is unclear how well AME-TS generalizes to more diverse downstream domains and forecasting settings.

(2) The regime predictor is central to the method, but its quality is not fully evaluated. The paper should report prediction accuracy or calibration for forecastability, seasonality, trend, and sparsity, and analyze how regime prediction errors affect forecasting performance.

(3) The descriptor choices are reasonable, but the paper should better justify why these four descriptors are sufficient. Other properties such as changepoints, noise level, intermittency type, exogenous dependence, or hierarchical structure may also be important for routing.

(4) The interpretability analysis is promising but could be more quantitative. t-SNE plots are helpful, but additional metrics such as mutual information between expert assignment and regime labels, expert load balance by regime, and seed-level stability would make the claim stronger.

Detailed Comments:

(1) Please clarify the computational overhead of the regime predictor and prior-alignment loss during training. The method is inference-efficient, but the training-time cost should also be reported.

(2) Please add more details on the validation performance of the regime predictor. Since the structural prior depends on predicted descriptors, this is important for reproducibility and trust in the method.

(3) Please include ablations at larger model scales. The current ablation table is useful, but it mainly reports AME-TS Tiny, so it is unclear whether the same trends hold for Base, Large, or Ultra models.

(4) Please discuss the M5 results more carefully. AME-TS performs especially well at lower hierarchy levels, but it is still weaker than the M5 winner at some higher aggregation levels. This distinction would help clarify when structure-guided MoE is most beneficial.

Justification of Score:

The paper presents a clear and technically meaningful idea: anchoring MoE routing with interpretable time-series descriptors to improve efficiency, interpretability, and adaptation stability. The results on GIFT-Eval and M5 are convincing, and the ablations support the value of structure-guided routing beyond sparse capacity alone. However, the paper would be stronger with broader downstream evaluation, more detailed analysis of the regime predictor, and more quantitative evidence for routing interpretability.